# Trophic Niche Overlap between Invasive and Indigenous Fish in a Northwest Reservoir of China

**Jie Wei** [1,2,3] **, Zhulan Nie** [3] **, Fenfen Ji** [1,2] **, Longhui Qiu** [1,2] **and Jianzhong Shen** [1,2,*]

1   Engineering Research Center of Green development for Conventional Aquatic Biological Industry in the Yangtze River Economic Belt, Ministry of Education, College of Fisheries, Huazhong Agricultural University, Wuhan 430070, China; weijiedky@126.com (J.W.); ffji1213266226@163.com (F.J.); longhui@webmail.hzau.edu.cn (L.Q.)
2   Key Laboratory of Freshwater Animal Breeding, Ministry of Agriculture, College of Fisheries, Huazhong Agricultural University, Wuhan 430070, China
3   Key Laboratory of Tarim Animal Husbandry Science & Technology, College of Animal Science, Tarim University, Alar 843300, China; niezhl2004@163.com
*   Correspondence: jzhsh@mail.hzau.edu.cn

**Abstract:** The Kizil reservoir in the Tarim River basin is an important habitat for the native Schizothoracinae fish (including *Aspiorhynchus laticeps*, *Schizothorax biddulphi*, *Schizothorax eurystomus*, *Schizothorax intermedius* and *Schizothorax barbatus*). Unfortunately, these species are threatened by many exotic fish, such as *Ctenopharyngodon idellus*, *Silurus asotus*. As an isolated habitat, the Kizil reservoir is an ideal area for studying biological invasions. However, the impact of invasive species on indigenous species in this reservoir remains unknown. In this study, the niche width and niche overlap between invasive and indigenous species in Kizil reservoir were studied based on stable isotope analysis. The results showed that niche width of two invasive species, *S. asotus* and *C. idellus*, was larger than that of native fish species, which confirmed the hypotheses that successful invaders have larger niche width. The niche overlap analysis showed that the two invasive species had high niche overlap with native fish species, which meant that there might be intensive interspecific competitions between them. The invasion of non-native species could be the main reason for the decrease of native species in the Kizil reservoir.

**Keywords:** Tarim River basin; Schizothoracinae fish; Kizil reservoir; biological invasion; stable isotope analysis; niche width; niche overlap

## 1. Introduction

Invasive species have strong impacts on ecosystem integrity and biodiversity. Freshwater ecosystems, especially rivers, which can be effectively considered biogeographic islands, are particularly susceptible to the establishment of invasive species because their ecological space is often 'unsaturated' with native species and is more likely to favor the establishment of non-natives [1,2]. Ecological effects of invaders may include behavioral shifts in native species, alteration of native habitat, alteration of food webs and trophic dependencies, and extirpation of native biota [3,4]. Two notorious freshwater invaders have established populations in most river systems of China in mainland Asia, and even other continents, i.e., *Ctenopharyngodon Idella* [5–7], and *Silurus asotus* [8–11], with potentially serious implications for native flora and fauna.

Quantifying the impacts of invasive species can be challenging due to the complexity of ecological interactions [12], particularly in aquatic ecosystems. Recently, stable isotope analysis (SIA) has been shown to be a useful tool for tracking changes in trophic structure and energy flows in an ecosystem, contributing to the further understanding of how an ecosystem may be affected by nonindigenous species [13,14]. Isotopic ratios are conserved up through the food web, with predictable isotopic shifts (or fractionation) at every trophic

step [15,16]. As such, stable carbon (information on food resources) and nitrogen infor-mation on trophic position) isotopic ratios can provide time-integrated information about feeding relationships and energy flow [17,18]. Thus, $\delta^{13}C$ and $\delta^{15}N$ values can be used to conceptualize trophic niches within communities and habitats because they vary both temporally and spatially [2,19,20] and provide a powerful approach to predict the inva-sion impacts of nonindigenous species and the degree of dietary competition on endemic species [21,22].

Innovative developments in isotope ecology have recently provided statistical Bayesian frameworks for investigating variation in isotopically defined groups [23,24]. Layman et al. (2007) proposed a series of quantitative stable isotope metrics to define the trophic ecology and structure of an ecosystem using $\delta^{13}C$ and $\delta^{15}N$ values [25]. Jackson et al. (2011) refined these techniques and bolstered their ability to cope with sample size disparities, allowing the degree of resource sharing (i.e., competition) between sympatric species to be quantified more easily [26].

Kizil Reservoir is an important habitat for five native schizothorax fish which were important economic fish in the Weigan river [27]. However, the Kizil reservoir has become a highly invaded ecosystem by numerous invasive species, including *Ctenopharyngodon idellus*, *Silurus asotus*, and so on [28]. These nonindigenous fish potentially threaten the persistence of native schizothorax fish through competitive or predatory interactions. It is disputed that invasive species might have endangered native schizothorax fish, because there is no conclusive evidence yet. Eggs of schizothorax fish are adhesive and are deposited over cobble and boulder substrate [29]. Once non-native fish interfere with their breeding area, it is very dangerous for the continuation of the population. In Lugu Lake, there have been precedents that small non-native fish (such as topmouth gudgeon,) prey on the eggs of native schizothorax fish [30]. *Aspiorhynchus laticeps* is a top predator in its range, but are at high risk of extinction. Among the many threats is competition for food from non-native carnivorous fish [31]. However, little is known about the trophic ecology of either native or non-native fish in the Kizil reservoir system. Using SIA and subsequent comparisons of species isotopic niche widths and their degree of overlap, our study aimed to prove that some invasive fish are likely to be directly competing with native fauna for dietary resources in the Kizil reservoir, and provide reference for specific measures to protect native biodiversity, thus providing support for the ecological conservation of the Kizil reservoir.

## 2. Materials and Methods

### 2.1. Study Area and Sampling

The XinJiang Weigan river is distributed in the basin between the north margin of Chaletag mountains and the south foothill of the middle part of Tianshan mountains in the northwest of China. It consists of five tributaries, including the Muzati river, the Kapuslang river, the Delevitchuk river, the Kalasu river and the Kizil river. Weigan river is an ideal area to study fish invasion because it is an inland river and isolated habitat. In 1985, Kizil Reservoir was built at the joining of the Muzati River and the Kizil River for the purpose of irrigation, flood control, and hydropower. It is the largest reservoir in XinJiang Uygur Autonomous Region, with a 16,637 thousand $km^2$ control catchment area and $6.4 \times 108$ cubic meters total capacity (Figure 1).

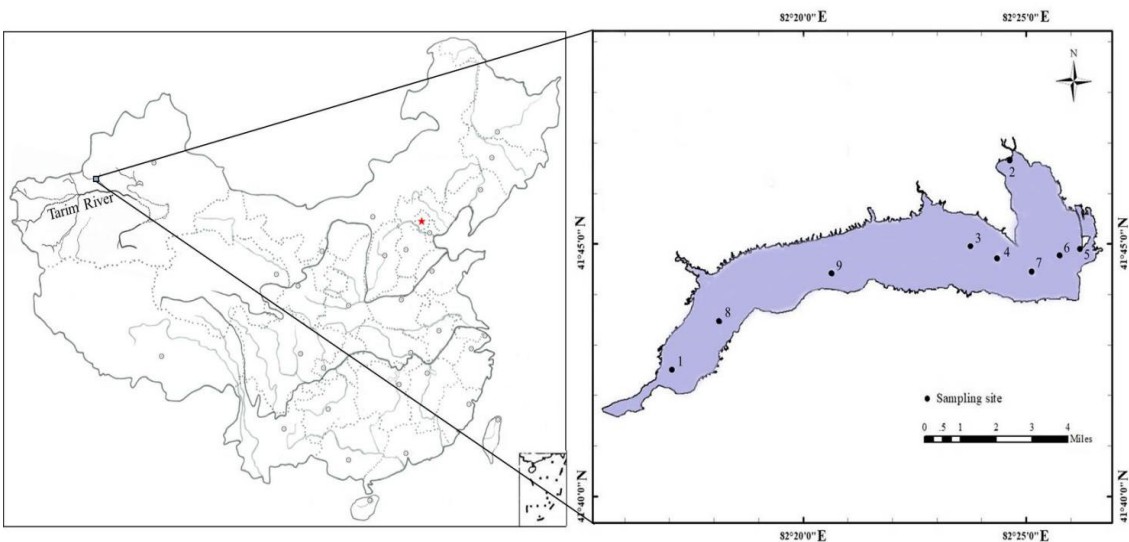

**Figure 1.** Locations of sampling sites in the Kizil Reservoir.

In this study, samples were collected at nine locations in the Kizil Reservoir (Figure 1). The sampling sites were located in the entrance to the reservoir of the Muzati river (site 1), the entrance to the reservoir of the Kizil river (site 2), the release sluice of the reservoir drain (site 5), and in pelagic zone of the reservoir (other sites). Sampling was conducted from upstream to downstream along the long-shaped reservoir seasonally in February, June, August and October of 2017–2018.

Environmental metrics were measured synchronously at the 9 locations in the Kizil Reservoir using a portable LAQUA act (HORIBA, Ltd., Kyoto, Japan), including depth, transparency (SD), temperature (T), dissolved oxygen (DO), conductivity (COND), salinity (SAL), total dissolved solids (TDS), and resistivity (RES) (Table 1). Total nitrogen (TN) and total phosphorus (TP) were analyzed based on a alkaline potassium persulfate digestion-UV spectro photometric method and ammonium molybdate spectrophotometric method, respectively [32,33].

**Table 1.** Physio-chemical characteristics (mean $\pm$ SD) in the Kizil Reservoir [a].

| Parameters | August | October | February | June |
|---|---|---|---|---|
| Depth (m) | 2.17 $\pm$ 1.61<br>n = 5 | 6.91 $\pm$ 3.72<br>n = 9 | 7.41 $\pm$ 4.66<br>n = 9 | 5.38 $\pm$ 3.96<br>n = 9 |
| SD (m) | 0.40 $\pm$ 0.34<br>n = 5 | 2.21 $\pm$ 1.12<br>n = 9 | 1.00 $\pm$ 0.26<br>n = 9 | 0.77 $\pm$ 0.61<br>n = 9 |
| pH | 7.68 $\pm$ 0.14<br>n = 5 | 8.79 $\pm$ 0.06<br>n = 9 | 7.99 $\pm$ 0.32<br>n = 9 | 8.73 $\pm$ 0.13<br>n = 9 |
| T (°C) | 23.94 $\pm$ 2.54<br>n = 5 | 14.72 $\pm$ 0.99<br>n = 9 | 0.34 $\pm$ 0.41<br>n = 9 | 22.23 $\pm$ 1.85<br>n = 9 |
| SAL (ppt) | - | - | 0.33 $\pm$ 0.04<br>n = 9 | 0.28 $\pm$ 0.04<br>n = 9 |
| DO (mg/L) | 7.43 $\pm$ 0.30<br>n = 5 | 10.26 $\pm$ 0.36<br>n = 9 | 9.86 $\pm$ 0.46<br>n = 9 | 7.52 $\pm$ 0.62<br>n = 9 |
| TDS (mg/L) | - | - | 447.47 $\pm$ 55.55<br>n = 9 | 377.66 $\pm$ 56.63<br>n = 9 |
| ORP (mv) | - | - | 157.06 $\pm$ 40.34<br>n = 9 | 173.16 $\pm$ 23.60<br>n = 9 |
| TN (mg/L) | 1.04 $\pm$ 0.51<br>n = 5 | 1.43 $\pm$ 0.16<br>n = 9 | 1.58 $\pm$ 0.19<br>n = 9 | 1.61 $\pm$ 0.23<br>n = 9 |
| TP (mg/L) | 0.06 $\pm$ 0.05<br>n = 6 | 0.02 $\pm$ 0.01<br>n = 9 | 0.05 $\pm$ 0.02<br>n = 9 | 0.03 $\pm$ 0.02<br>n = 9 |

[a]: SD: transparency; T: temperature; COND: conductivity; SAL: salinity; RES: resistivity; TDS: total dissolved solids; DO: dissolved oxygen; ORP: oxidation-reduction potential; TN: total nitrogen; TP: total phosphorus.

### 2.2. Sample Analysis

The leaves of hygrophyte were collected by hand and washed with distilled water to remove detritus. Submerged plants, free-floating plants and floating-leaved plants were not found in the Reservoir. Phy-toplankton samples were obtained from hauls of a 64 μm Phytoplankton net and were filtered through precombusted Whatman GF/F filters (450 °C for 6 h). Particulate organic matter (POM) was collected by filtering mixing water (from the upper layer, middle layer and lower layer of water) that was passed through a 64 μm phytoplankton net onto pre-combusted GF/F filters, under moderate vacuum (10 mbar), until clogging [34,35].

Zooplankton were sampled with a zooplankton net (112 μm mesh size) repeatedly in the open water for approximately 15–20 min, while benthic invertebrates were collected with a Peterson grab (1/16 m$^2$). Both zooplankton and benthic invertebrates were then held in water for 24 h to void their guts [36].

Fish sampling was performed using a cast net (10–16 mm), gill net (18–45 mm), and fyke net to cover all habitat types because each station has different characteristics such as depth (0.42 to 12.7 m) and transparency (0.05–3.6 m). Fish tissue was extracted from the white dorsal muscle in the field. All the sampled fish were identified, enumerated, measured (total length) and weighed.

All samples were dried in an oven at 60 °C and grounded into fine powders using a mortar and pestle. Prior to $\delta^{13}$C measuring, POM, phytoplankton, zooplankton were treated with 1 N HCl to eliminate the carbonates using a drop-by-drop technique [37]. All samples for $\delta^{15}$N analyses did not undergo any treatment of acidification. Each treated sample was then dried again at 60 °C, and grounded into fine powders [38].

### 2.3. Stable-Isotope Analysis

All stable isotopic samples were analyzed at the Key Laboratory of Crop Ecophysiology and Farming System for the Middle Reach of the Yangze River, Ministry of Agriculture using vario ISOTOPE cube elemental analyzer coupled to an Isoprime-100 isotope mass spectrometer. The stable isotopic ratios were expressed in the conventional "δ" notation as parts per thousand (‰) according to the following equation [34]:

$$\Delta R = [(X \text{ sample} - X \text{ standard})/X \text{ standard}] \times 10^3 \ (‰) \tag{1}$$

where R is $^{13}$C or $^{15}$N and X is $^{13}$C/$^{12}$C or $^{15}$N/$^{14}$N. Atmospheric nitrogen (for $\delta^{15}$N) and Peedee belemnite (PDB) (for $\delta^{13}$C) were used as the standards. The analytical precision was within 0.1‰ and 0.2‰ for carbon and nitrogen isotope measurements, respectively.

The following formula was used for the calculation of relative trophic level (TL):

$$TL = \frac{\delta^{15}N_{consumer} - \delta^{15}N_{baseline}}{3.4} + 2 \tag{2}$$

where 3.4 is the assumed enrichment in $\delta^{15}$N between successive TL, which has been identified as an average trophic nitrogen fractionation for aquatic consumers [39]. *Chironomus salinarius* was used as the food-web baseline indicator in our study (TL = 2) because they were abundant at all sites, and they were frequently found in the diets of most fish species, especially the omnivorous Schizothorax fish.

### 2.4. Statistical Analysis

Statistical analyses were conducted using the programs SPSS 19.0. Kruskal-Wallis tests were used to test for differences in $\delta^{13}$C and $\delta^{15}$N signatures among the basal food sources. Particularly, the Mann-Whitney U test was executed to compare the difference between the isotopic signatures of any two sources/consumers and comparisons of density, biomass of benthic invertebrates, phytoplankton. Before the analyses, the data were checked for normality and homogeneity of variance assumptions and logarithmic transformations were performed when needed. All tests maintained a comparison-wise type I error rate of 0.05.

δ¹³C and δ¹⁵N values were pooled for each species to derive quantitative population metrics [25], and investigate trophic structure and dietary resource competition between the invasive and indigenous fauna in the Kizil reservoir ecosystem. Fish muscle tissues displayed high C:N ratios (>3.5, Table S1), indicating they contained a high lipid content, which can bias analyses. To counter this potential bias, we corrected for lipid content using Post's method [40]. Metrics and analysis were conducted using the Stable Isotope Bayesian Ellipses in an R model [26]. Metrics included nitrogen (dNrb) and carbon (dCrb) ranges, providing a univariate measure of the total nitrogen and carbon ranges exploited by a species; mean distance to the centroid (CDb), which provides a description of trophic diversity; standard deviation of nearest neighbor distance (SDNNDb), which provides a measure of trophic evenness; and standard ellipse area (SEAc); which provides a bivariate measure of mean core isotopic niche [26]. For detailed methodology and original descriptions of stable isotope metrics, please refer to previous published literature [25,26]. The calculation of SEAc allows for a measure of the degree of niche overlap (100% indicating complete overlap), which can then be used as a quantitative measure of dietary similarity between species and was quantified according to previously described methods [41]. To allow comparisons between species with varying sample sizes, all metrics were bootstrapped (n = 10,000, indicated with subscript 'b') based on the minimum consumer sample size in the data set. A small sample size correction for improving accuracy to SEA values is indicated by the subscript 'c' [26].

Both SIBER and SIAR models were run in the R environment (R Development Core Team, 2007; available at http://cran.r-project.org/web/packages/siar/index.html (accessed on 10 August 2021).

## 3. Results

In this study, 36 types of samples were collected in the Kizil reservoir (Figure 2). Among them, 17 species of fish, two species of invertebrates and nine species of hygrophyt can be identified. According to δ¹⁵N values, the samples in the Kizil Reservoir are roughly divided into three groups: the higher-level consumer groups including *Abbottina rivularis*, *Aspiorhynchus laticeps*, *Carassius auratus*, *Ctenogobius giurinus*, *Cyprinus carpio*, *Hemicculter Leuciclus*, *Hypseleotris Swinhonis*, *Pseudorasbora parva*, *Rhinogobius giurinus*, *Schizothorax eurystomus*, *Silurus asotus*, *Triplophysa teunis*, *Triplophysa teunis*, *Exopalaemon carinicauda*, *Macrobrachium nipponense*, uncertain shrimp; the lower-level consumer group including Chironomid larvae, *Ctenopharyngodon idella*, *Schizothorax barbatus*, *Schizothorax biddulphi*, *Schizothorax intermedius*; the producer-based group includes nutrient sources and zooplankton. According to δ¹³C values, zooplankton, phytoplankton, POM, *Bolboschoenus yagara* and unknown hygrophyt are most likely to become food sources for more advanced consumers.

Stable isotope metrics showed that the non-native carnivorous fish *S. asotus* occupied a larger trophic niche than five native Schizothorax fish. The larger ranges of NRb (5.36) for the invasive fish indicated that they had greater degree of trophic diversity than that of indigenous fish (Table 2). The medium ranges of CRb (1.87) for invasive fish showed that they utilized more multiple basal resources with varying δ¹⁵N values than the indigenous fish, with the exception of *S. biddulphi* and *A. laticeps*. A medium CD value (1.17) for the invasive fish described medium trophic diversity, with three native species possessing higher diversity and two possessing lower diversity (Table 3). Concurrently, the medium ranges of SDNND values of the non-native fish were much higher than those of most indigenous fish, with the exception of *A. laticeps* (Table 2).

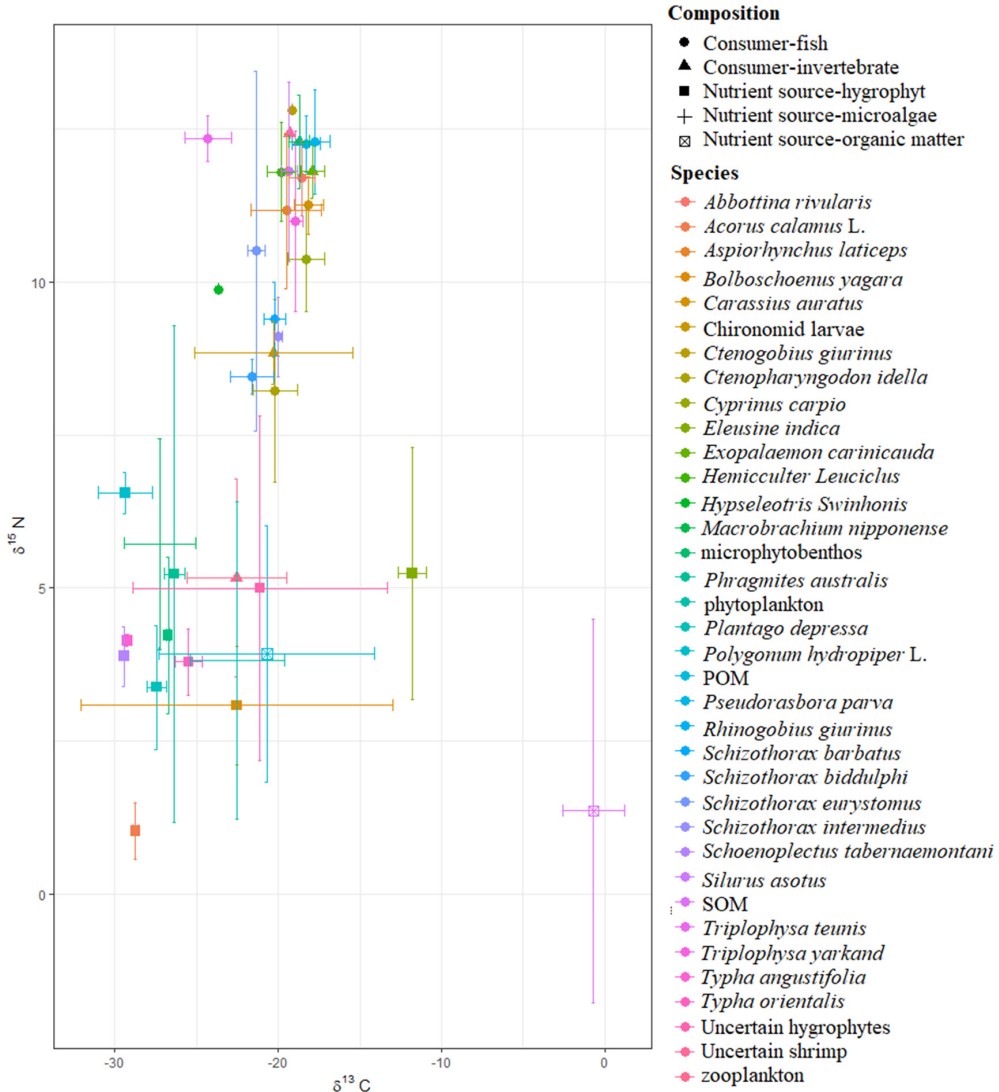

**Figure 2.** Overall trophic position based on mean stable isotope values (error bars indicate standard deviation) for organisms from the Kizil reservoir. (POM: particulate organic matter, SOM: soil organic matter).

**Table 2.** Stable isotope metrics for 13 fish species in the Kizil Reservoir [a].

| Species | n | TL | NRb | CRb | CD | MNND | SDNND | TA | SEAc |
|---|---|---|---|---|---|---|---|---|---|
| Native species | | | | | | | | | |
| *Aspiorhynchus laticeps* | 6 | 2.69 ± 0.37 | 3.90 | 5.77 | 1.71 | 1.24 | 1.83 | 4.94 | 5.20 |
| *Schizothorax biddulphi* | 5 | 1.89 ± 0.08 | 0.75 | 2.83 | 1.19 | 0.24 | 0.31 | 0.94 | 1.55 |
| *Schizothorax eurystomus* | 4 | 2.49 ± 0.86 | 5.11 | 1.09 | 2.58 | 0.03 | 0.02 | 1.31 | 1.67 |
| *Schizothorax intermedius* | 4 | 2.08 ± 0.19 | 1.22 | 0.48 | 0.57 | 0.13 | 0.02 | 0.17 | 0.25 |
| *Schizothorax barbatus* | 7 | 2.16 ± 0.18 | 1.35 | 1.96 | 0.72 | 0.25 | 0.42 | 1.32 | 1.53 |
| Non-native species | | | | | | | | | |
| *Silurus asotus* | 10 | 2.87 ± 0.43 | 5.36 | 1.87 | 1.17 | 0.65 | 0.54 | 4.11 | 1.92 |
| *Ctenopharynagodon idellus* | 11 | 1.82 ± 0.44 | 4.74 | 3.70 | 1.82 | 0.82 | 0.50 | 10.71 | 6.57 |
| *Rhinogobius giurinus* | 11 | 3.00 ± 0.13 | 1.24 | 3.19 | 0.81 | 0.44 | 0.57 | 2.17 | 1.38 |
| *Pseuderasbora parva* | 12 | 3.02 ± 0.25 | 2.62 | 3.46 | 0.98 | 0.55 | 0.26 | 3.58 | 1.69 |
| *Cyprinus carpio* | 27 | 2.45 ± 0.25 | 3.94 | 4.98 | 1.18 | 0.41 | 0.30 | 11.36 | 3.05 |
| *Carassius auratus* | 24 | 2.71 ± 0.14 | 1.50 | 3.71 | 0.86 | 0.28 | 0.22 | 3.25 | 1.15 |
| *Hemiculter leacisculus* | 12 | 2.87 ± 0.24 | 3.13 | 2.69 | 0.83 | 0.48 | 0.47 | 4.08 | 1.96 |
| *Abbottina rivularis* | 7 | 2.84 ± 0.18 | 1.97 | 2.24 | 0.87 | 0.79 | 0.12 | 2.48 | 1.85 |

[a]: TL: trophic level; NR: d15N Range; CR: d13C range; CD: mean distance to centroid; MNND: Mean nearest neighbor distance; SDNND: Standard deviation of nearest neighbor distance.

**Table 3.** The proportion (%) of SEAc overlap between fish species in the Kezil reservoir [a].

| Species | A. laticeps | S. biddulphi | S. eurystomus | S. intermedius | S. barbatus | S. asotus | C. idellus | R. giurinus | P. parva | C. carpio | C. auratus | H. leacisculus | A. rivularis |
|---|---|---|---|---|---|---|---|---|---|---|---|---|---|
| **A. laticeps** | | 3 | 1 | 1 | 15 | 39 | 42 | 31 | 17 | 51 | 30 | 41 | 38 |
| **S.biddulphi** | 17 | | 1 | 2 | 28 | 7 | 76 | 0 | 0 | 9 | 0 | 0 | 0 |
| **S. eurystomus** | 29 | 11 | | 0 | 23 | 2 | 46 | 1 | 0 | 5 | 0 | 14 | 1 |
| **S.intermedius** | 25 | 35 | 0 | | 88 | 53 | 98 | 0 | 0 | 60 | 0 | 10 | 1 |
| **S. barbatus** | 40 | 24 | 1 | 8 | | 47 | 96 | 0 | 0 | 61 | 1 | 15 | 2 |
| S. asotus | 59 | 2 | 0 | 1 | 16 | | 43 | 43 | 22 | 55 | 36 | 63 | 52 |
| C.idellus | 14 | 19 | 1 | 2 | 24 | 18 | | 2 | 1 | 32 | 4 | 5 | 5 |
| R. giurinus | 74 | 0 | 0 | 0 | 0 | 53 | 17 | | 62 | 44 | 34 | 49 | 69 |
| P. parva | 58 | 0 | 0 | 0 | 0 | 31 | 14 | 70 | | 47 | 27 | 23 | 49 |
| C. carpio | 40 | 3 | 0 | 1 | 15 | 34 | 71 | 13 | 12 | | 40 | 14 | 37 |
| C. auratus | 68 | 0 | 0 | 0 | 2 | 40 | 46 | 42 | 28 | 92 | | 30 | 80 |
| H. leacisculus | 68 | 0 | 0 | 1 | 9 | 70 | 32 | 48 | 12 | 42 | 40 | | 55 |
| A. rivularis | 79 | 0 | 0 | 0 | 2 | 61 | 37 | 66 | 32 | 75 | 73 | 51 | |

[a]: Nutritional niche overlap contains two directions [24,26], the number in the table means niche overlap percentage of longitudinal species to horizontal species. The species in bold lines are native species, and the others are non-native species.

Comparison of stable isotope metrics between the other non-native herbivorous fish *C.idellus* and native fish showed similar NR and CR ranges. The larger ranges of NR (4.74) and CR (3.70) for *C.idellus* indicated that it occupied a larger niche with a more diverse diet than three native Schizothorax fish (*S. biddulphi*, *S. intermedius* and *S. barbatus*), but less than two native fish species (*S. eurystomus* and *A. laticeps*) (Table 2). This was consistent with the SEAc value for *C. idellus* (6.57) (Table 2), but was contrary to the lower trophic level value for *C. carpio* (Table 2). The TA value of the invasive herbivorous fish (10.71) was much higher than for native fish (Table 2).

The species in bold lines are native species, and the others are invasive species. The niche overlap analyses showed that there was a medium degree of overlap between the invasive carnivorous fish *S. asotus* and indigenous carnivorous fish *A. laticeps* (Table 3). The proportion of SEAc overlap between *A. laticeps* and *S. asotus* was 39%, and that between *S. asotus* and *A. laticeps* was 59%. Niche overlaps between the invasive fish and native omnivorous counterparts were very low, with the majority of shared food resources having overlaps less than 16%. The proportion of SEAc overlap between *S. asotus* and *S. intermedius* and between *S. asotus* and *S. barbatu* was 53% and 47% respectively.

The niche overlap analyses showed that there was a greater degree of trophic overlap between herbivorous fish *C. idellus* and native omnivorous schizothorax fish (Table 3). The proportion of SEAc overlap between *C. idellus* and *S. intermedius* was 98%, 96% between *C. idellus* and *S. barbatus*, and 76% between *C. idellus* and *S. biddulphi*. Additionally, there was 42% of the A. laticeps individual distributed in *C. idellus* niche space. However, the niche overlap value of the invasive fish to five native schizothorax fish was very low, with the majority of shared food resources having overlaps of less than 24%.

SIAR mixing models showed that five native schizothorax fish and two invasive fish shared the primary food source with similar contributions (Figure 3). Phytoplankton and microphytobenthos contributed the highest proportion to these fish diets (0.29–0.38), followed by POM (0.24–0.29) and hygrophyt (0.19–0.28). The proportion of sediment was low (0.09–0.19) (Figure 2).

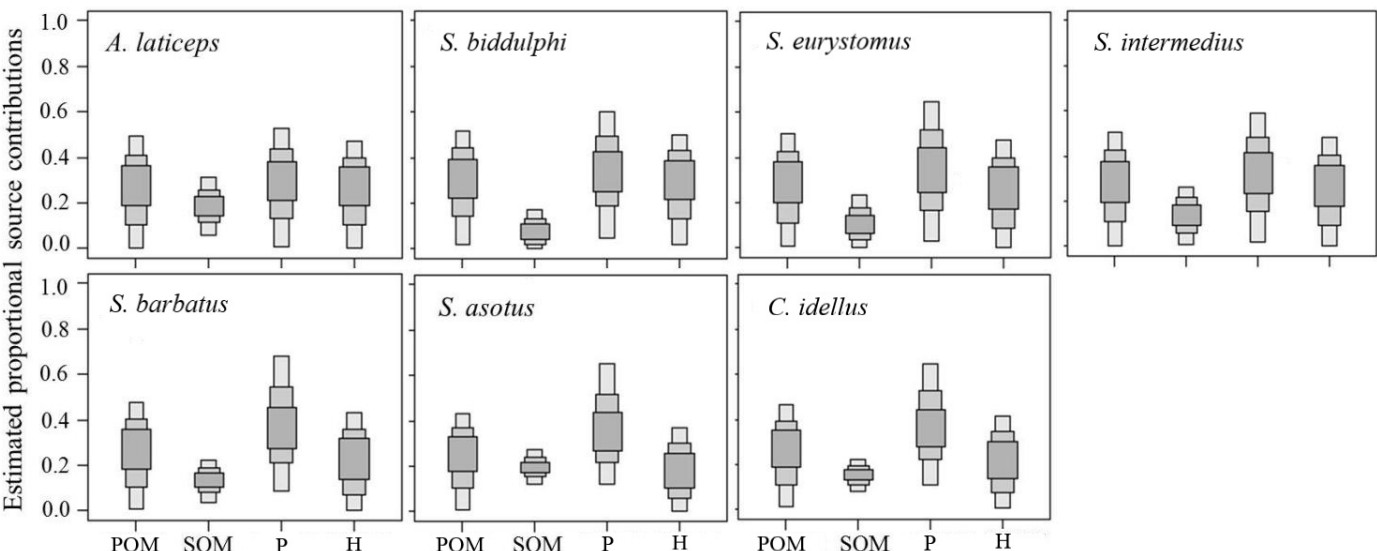

**Figure 3.** Estimated proportional source contributions (with 50%, 75% and 95% credibility intervals; POM: particulate organic matter, SOM: soil organic matter, P: phytoplankton and microphytobenthos, H: hygrophyt) as determined by SIAR [42] for five native schizothorax fish (*A. laticeps*, *S. biddulphi*, *S. eurystomus*, *S. intermedius*, *S. barbatus*) and two invasive fish (*S. asotus*, *C.idellus*).

## 4. Discussion

The isotopic niche is a useful tool to examine the competition and potential impact of invasive species on native ones. The variance in the stable isotope values of species can be

used as an indicator of feeding niche widths [43]. The $\delta^{13}$C values are mainly controlled by carbon sources, and the fractionation of biological metabolism has little effect. From primary producer to primary consumer, $\delta^{13}$C characteristics remain largely unchanged, so it can be used to indicate and track the origin of food and to calculate the contribution of each food source [44]. The $\delta^{15}$N values of consumers are generally 3.4 $\pm$ 1.1 higher than those of their food, and this value is relatively stable. Therefore, $\delta^{15}$N is more suitable for the evaluation of trophic level [45,46]. In this study, biplots were used to show the trophic positions of all members of the Kizil reservoir, focusing on the niche width, niche overlap and nutrient source contributions of native and non-native fish. The results showed that *S. Asotus* and *C. Idellus* had intensive interspecific competitions with native fish.

The quantitative structure of populations in the ecosystem can most directly reflect biological invasion [47]. In this study, more than 70% of the fish caught were non-native fish. Non-native fish have an absolute advantage in population numbers. A survey of fish resources in the Weigan River in 2019 showed that non-native fish such as *Perca fluviatilis* and *P. parva* not only have a quantitative advantage in the middle and lower reaches of the Weigan River, but most of them have different sizes individuals. Some of them have sexual maturity, which indicates that the non-native fish have adapted and successfully established a colony and spread [48]. It can be seen that the native fish and non-native fish in the Kizil reservoir already have the basic conditions for interspecies competition in terms of population structure. It is well-known that interspecies competition is an important factor to determine the trophic niche width of coexisting species [49,50]. The non-native species with width niche and high trophic overlap will reduce the trophic niches of native species [51], especially in the nutrient-poor Kizil reservoir, which meant that those non-native species have greater advantage in habitat use than the native schizothorax fish. Those five schizothorax fish species are less abundant, and the fact that these rare species sizes are small may reflect competitive exclusion [52]. The impact of those non-native species on native fish is clear, while taking the vulnerability of those five schizothorax fish species into consideration.

Conflict feeding behavior is one of the important factors leading to interspecific competition [53–55]. Among the fish samples collected in this study, there are two kinds of carnivorous fish, which are the native fish *A. laticeps* and the non-native fish *S. asotus*. It has been recorded that before biological invasion in Xinjiang water areas, *A. laticeps* was the top consumer in the ecosystem [31], but in this study, its TL was lower than that of *S. asotus*, and even two omnivorous fish, *R. giurinus* and *P. parva*. This may be due to the increasing number of non-native fish that put the native fish at a disadvantage in the com-petition for food. The higher TL of non-native omnivorous fish may be because they prey on the eggs or fry of other fish and interfere with reproduction [56,57]. However, the occurrence of this situation cannot be confirmed only from stable isotopes analysis. Observation of feeding behavior and the identification of the contents of the digestive tract is necessary to combine stable isotope analysis to make an accurate judgment. Among the omnivorous fish, *C.idellus* has a high niche overlap with the four native omnivorous schizothorax fish. High trophic overlap suggested that either food resources are not limiting, or niches are not partitioned well between these species. Although *C.idellus* are widely known as herbivorous fish, they are carnivorous in their early stages of life, feeding on zooplankton and benthos [58], which may well explain their broad trophic niche. Those invasive species occupied a vacant niche space particularly in unfavorable environmental conditions for native fish, which meant that there might be intensive interspecific competitions between them and native species.

In conclusion, the high niche overlap between those two invasive species is probably one of main reasons for decline in schizothorax fish resources in the Kizil reservoir and Weigan river. This study presented some explanatory arguments on the possible impacts of invasive species on native fish species in a dynamic ecosystem. Stable isotope metrics and the niche overlap analysis showed that those two invasive species, *S. asotus* and *C. idellus*, had larger niche width than native schizothorax fish species. It confirmed the hypotheses that successful invaders have a large niche width [59,60]. Except for typical advantages for

successful invasion, the wide isotopic niche area and competition ability of *S. asotus* and *C. idellus* was assessed in this study. It is clear that those two invasive species have important functional roles in the community dynamics, with high dominance, high niche area and high degree of niche overlap with those five schizothorax fish species. We suggest that the different microhabitat characteristics in the reservoir and Weigan river ecosystem should be maintained and long-term monitoring is needed for the river management plan.

**Supplementary Materials:** The following are available online at https://www.mdpi.com/article/10.3390/w13233459/s1, Table S1: The C:N ratio for 13 fish species in the Kizil Reservoir.

**Author Contributions:** Conceptualization, J.W. and J.S.; methodology, J.W.; software, Z.N. and F.J.; validation, J.S.; formal analysis, L.Q.; investigation, Z.N.; resources, J.W.; data curation, Z.N. and L.Q.; writing—original draft preparation, J.W.; writing—review and editing, J.S. and F.J.; visualization, Z.N.; supervision, J.S.; project administration, J.W.; funding acquisition, J.S. All authors have read and agreed to the published version of the manuscript.

**Funding:** This research was funded by the National Natural Science Foundation of China, grant number 31860729 and 31560721, United Fund of Huazhong Agricultural University and Tarim University, grant number 2662017PY118 and TDHNLH201702, United Fund of the Ocean University of China and Tarim University, grant number ZHYLH201902, The Young and Middle-aged Science and Technology Innovation Leading Talent Program Project of Xinjiang Production and Construction Corps, grant number 2018CB033.

**Institutional Review Board Statement:** The study was conducted according to the guidelines of the Declaration of Helsinki, and approved by the Scientific Ethic Committee of Huazhong Agricultural University (SYXK2017-0002; 2017-1-18).

**Informed Consent Statement:** Not applicable.

**Data Availability Statement:** The data presented in this study are available on request from the corresponding author. The data are not publicly available due to they involve other unpublished studies.

**Acknowledgments:** We thank Kizil Reservoir Administration Bureau of Xinjiang Water Conservancy Department for help sampling with legal permissions. We thank the Key Laboratory of Crop Eco-physiology and Farming System for the Middle Reach of the Yangze River (Huazhong Agricultural University) for providing technical assistance on sample analysis. We thank the project team (Xiong Lei, Sun Xiuhua, Sun lindan, Kang Zhipeng and Zhou Mengxia) for valuable support in the field studies and laboratory analyses.

**Conflicts of Interest:** The authors declare no conflict of interest.

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
