# Peer review of "Trophic Niche Overlap between Invasive and Indigenous Fish in a Northwest Reservoir of China"

_water, doi:10.3390/w13233459_

Round 1
Reviewer 1 Report
Jie and colleagues report on a stable isotope analysis of invasive and indigenous fishes in an impoundment. While the authors did not include explicit objectives, the authors collected tissue samples from a variety of native and non-native fishes, ultimately finding that many fishes have overlapping trophic niches and, in particular, two non-native fishes have broad trophic niches.
The article is mostly well written, and, overall, I found the stable isotope approach to be mostly sound, although I do note some specific issues in my specific comments below. Further, there are a few issues (some major) that might need to be addressed (if possible, or, at least explained). For example, the authors devote too little space to discussing the actual species and their habits. Ultimately, I think the article could be suitable for publication is the authors do a careful revision, focusing on specific hypotheses, focusing on the particular fish species, focusing on the limitations of stable isotopes used singularly, and explicitly tying the results to the research that has already been conducted.
Major Issues
- In the last part of the introduction the authors include an extensive section on the study system; this section of the introduction is followed by a section in the Materials and Methods on Study area and sampling. I would suggest the authors move some of this material on the study system to the methods. Further, the life history of these fishes is almost entirely missing. This last section of the introduction should be used to discuss the non-native (and native) fishes, communicating diet, in particular. Ultimately, the authors should include a hypothesis instead of stating a goal of investigating whether invasive fishes compete with native fishes.
- The authors should include a more thorough communication of competition, how it is recognized in natural settings, and how stable isotope analysis can be used to infer apparent competition. The authors should also discuss the limitations of stable isotope analysis. As is, the authors infer competition, but I don’t think it is that convincing. Simply showing niche overlap is insufficient to conclude competition is taking place. For example, good evidence might have stable isotope samples prior to invasion or even using uninvaded sections as controls.
- The discussion needs to be revised. It is truncated, fails to connect the stable isotope results to the actual life history and diets of the species, and is poorly cited.
Specific Comments
L10: insert “the” between “in Tarim”
L12: replace “were” with “are”.
L13: remove “and so on”. This is an informal phrase that doesn’t communicate any information.
L69-70: It seems this statement that is disputed needs a citation.
L88-89: Why did the authors sample seasonally, from February to October? Are there any drawbacks to collecting samples across seasons?
L94-95: It is not clear what these protocols refer to; is there a reference to a TN TP protocol?
L102: Not clear what the difference is between floating plants and leave floating plants. Further, if there are no submerged plants, what are the grass carp feeding on?
L103-105: This sample would not be just phytoplankton; there would be rotifers caught in the plankton net: rotifers, copepods, cladocerans. Ultimately, it is not clear how these samples (and others, benthic inverts, zoops, etc.) were utilized as they are not again mentioned in the manuscript.
L118-122: Samples were not corrected for potential variation in lipids. The authors should indicate whether this was necessary.
Post, D. M., C. a Layman, D. A. Arrington, G. Takimoto, J. Quattrochi, & C. G. Montaña, 2007. Getting to the fat of the matter: models, methods and assumptions for dealing with lipids in stable isotope analyses. Oecologia 152: 179–189
L129: Check the spelling for “parts per mile”
L137-138: The authors should indicate that they are not calculating absolute trophic position; instead, this is just a relative position based on the chosen baseline.
Table 3: At first glance it is not clear from the table where the native species end and the invasive species (I prefer the term non-native) begin.
Figure 2. Not clear what the abbreviations are on the x axes. Include terms in the figure caption.
L229-238: This is a reiteration of the methods used; delete this section and begin with your most important results.
L253-254: how does a fish that feeds upon submerged vegetation have a broad trophic niche; the authors should go into some detail about their results.
L259: spell out 5
Reviewer 2 Report
Review
Trophic niche overlap between invasive and indigenous fishes in a northwest reservoir of China
I found the present work a greater contribution. Unfortunately, some standard procedures in marine tissues have been omitted or not explained in M&M. My main source of concern is related to the methodology. The Authors should explain or detail the methods in depth. So, if non standard procedures were applied in samples with N:C rations >3.4, the present results are biased.
More comments and explanations:
Introduction.
Authors have introduced the study properly.
Material and methods
I found the reduced number of individuals and the unbalanced number of samples within a species a motive of alarm. However, the authors justified these circumstances during the discussion. In my opinion, a balanced sampling guarantees higher quality and credibility on the results.
As far as I know, some studies indicate that mathematical normalization is sufficient to account for bias in δ(13) C values associated with lipid content in fish tissues when C:N ratios are above 3.5. But, C:N ratios below 3.5 indicate that tissues have insufficient levels of lipid to bias the δ(13) C values. Generally, these findings support the use of more timely and cost-effective processing and analysis methods in future aquatic food-web studies utilizing SIA. Please clarify and include these aspects in the methodology and possible biases, just in case. Where are presented the C:N ratio?
Here an example of standard procedure:
“Fish muscle tissues displayed high C:N ratios (> 3.5) indicating they contained a high lipid content, which can bias analyses. To counter this potential bias, we corrected for lipid content using the Fry (FMB) method. “
Results
Based on the great amount of food-web analysed during the surveys in the present study, I missed the typical plot explaining the trophic position of all the species sampled. I encourage the authors to include a Stable isotope biplots (δ13C and δ15N) showing the food web the habitat.
Regards
Round 2
Reviewer 2 Report
The authors have adessed all my concern and answered point by point all the questions.
Regards.